# Mitochondrial Pseudogenes Suggest Repeated Inter-Species Hybridization among Direct Human Ancestors

**DOI:** 10.3390/genes13050810

**Published:** 2022-05-01

**Authors:** Konstantin Popadin, Konstantin Gunbin, Leonid Peshkin, Sofia Annis, Zoe Fleischmann, Melissa Franco, Yevgenya Kraytsberg, Natalya Markuzon, Rebecca R. Ackermann, Konstantin Khrapko

**Affiliations:** 1School of Life Sciences, École Polytechnique Fédérale de Lausanne, 1015 Lausanne, Switzerland; konstantinpopadin@gmail.com; 2Center for Mitochondrial Functional Genomics, Institute of Living Systems, Immanuel Kant Baltic Federal University, 236040 Kaliningrad, Russia; 3Swiss Institute of Bioinformatics, 1015 Lausanne, Switzerland; 4Institut Gustave Roussy, 94805 Villejuif, France; genkvg@gmail.com; 5Department of Genetics, Harvard Medical School, Boston, MA 02115, USA; peshkin@gmail.com; 6Department of Biology, Northeastern University, Boston, MA 02115, USA; annis.s@northeastern.edu (S.A.); fleischmann.z@northeastern.edu (Z.F.); franco.mel@northeastern.edu (M.F.); 7BIDMC and Harvard Medical School, Boston, MA 02215, USA; yevgenya.kraytsberg@gmail.com; 8Draper Laboratory, Cambridge, MA 02139, USA; nmarkuzon@yahoo.com; 9Human Evolution Research Institute, Department of Archaeology, University of Cape Town, Cape Town 7700, South Africa; becky.ackermann@uct.ac.za

**Keywords:** human evolution, hybridization, mtDNA, NUMT

## Abstract

The hypothesis that the evolution of humans involves hybridization between diverged species has been actively debated in recent years. We present the following novel evidence in support of this hypothesis: the analysis of nuclear pseudogenes of mtDNA (“NUMTs”). NUMTs are considered “mtDNA fossils” as they preserve sequences of ancient mtDNA and thus carry unique information about ancestral populations. Our comparison of a NUMT sequence shared by humans, chimpanzees, and gorillas with their mtDNAs implies that, around the time of divergence between humans and chimpanzees, our evolutionary history involved the interbreeding of individuals whose mtDNA had diverged as much as ~4.5 Myr prior. This large divergence suggests a distant interspecies hybridization. Additionally, analysis of two other NUMTs suggests that such events occur repeatedly. Our findings suggest a complex pattern of speciation in primate/human ancestors and provide one potential explanation for the mosaic nature of fossil morphology found at the emergence of the hominin lineage. A preliminary version of this manuscript was uploaded to the preprint server BioRxiv in 2017 (10.1101/134502).

## 1. Introduction

Increasingly, the emergence and evolution of our species are being revealed as a period characterized by genetic exchange between divergent lineages. For example, we now have evidence of hybridization between Neanderthals and early humans originating from Africa, between Denisovans and early humans, between Neanderthals and Denisovans, and between Denisovans and an unidentified hominin [1,2,3,4]. There is also evidence of a first-generation child of a Denisovan father and a Neanderthal mother [5] and of genetic exchange between recent and ancient lineages in Africa [6,7,8,9,10,11]. These studies indicate that hybridization was prevalent during the period of the emergence of *Homo sapiens* and suggest that it may be the rule rather than the exception in hominin evolution. However, we have little information on the presence or prevalence of hybridization during earlier (pre-1 Ma) periods in human evolution. In 2006, a study by Patterson and colleagues [12] concluded that the hominin lineage first significantly diverged from the chimpanzee lineage but later hybridized back before finally diverging again. This study prompted an intense debate [13,14,15,16,17,18] but has remained the sole piece of evidence for such an early admixture event. Although none of the subsequent studies fully rejected or confirmed the hybridization scenario, most pointed to the lack of sufficient evidence to uphold it. The research presented here supports a similar early hybridization scenario using an entirely different approach. Moreover, our analyses suggest that interbreeding occurred repeatedly among our distant ancestors.

*Hominin* here denotes a human lineage upon its separation from the chimpanzee. The term *Hominine*, in contrast, denotes the human, chimpanzee, and gorilla clade, upon its separation from the orangutan lineage.

The evidence for interspecies hybridization presented here comes from a special type of pseudogenes (“NUMTs”) that are fragments of mitochondrial DNA (mtDNA) integrated into the nuclear genome. There are hundreds of NUMT sequences in the human genome [19]. The NUMTs found in the present-day human genome have been inserted into nuclear DNA since tens of millions of years ago and this process continues today (Srinivasainagendra 2017). We have recently reported evidence that the insertion of NUMTs into the nuclear genome might have accelerated during the emergence of the genus *Homo* [20]. NUMTs are considered “DNA fossils” since they preserve ancient mtDNA sequences virtually unchanged due to a significantly lower mutational rate in the nuclear versus mitochondrial genome [21]. NUMTs, therefore, offer an opportunity to peek into the distant past of populations [22,23,24]. Here, we demonstrate that a NUMT on chromosome 5 descends from a mitochondrial genome that had been highly divergent from our ancestors’ mtDNA at the time of becoming a pseudogene. This implies that this pseudogene should have been created in an individual from a (hominine) species that at the time of insertion was highly diverged from our direct ancestor. For this pseudogene to end up in our genome, this (now extinct) hominine should have hybridized with our direct ancestors. Moreover, our analysis of additional NUMTs with similar phylogenic history implies that this scenario was not unique.

## 2. Results

### 2.1. A NUMT on Chromosome 5 Originated from a Highly Divergent Mitochondrial Genome 

In an early screen of human pseudogenes of mtDNA [25], we discovered a pseudogene sequence on chromosome 5, which later turned out to be a large (~9 kb) NUMT, referred to here as “ps5”. We then discovered close homologs of ps5 in the chimpanzee and gorilla genomes, i.e., in all contemporary *hominines*. Ps5 was absent in orangutans and more distant primates (see Appendix A). This NUMT turned out to have an extraordinary evolutionary history.

A joint phylogenetic tree of the three *hominine* ps5 NUMTs and the mtDNA sequences of great apes (Figure 1A) has a very surprising shape. One would expect that, as selectively neutral loci, pseudogenes should approximately follow the evolutionary paths of the species in which they reside. That is, the NUMT sub-tree should resemble the mtDNA sub-tree, which is a good representation of ape/human evolution. One may expect, though, that all branches of the NUMT tree should be shorter than those of the mtDNA tree, as the mutation rate in nuclear DNA is expected to be lower than in mtDNA. Contrary to these expectations, the NUMT has the following very different shapes: a very long stem (“ps5 stem”) and short branches. Even more intriguingly, phylogenetic mutational analysis (Appendix A) showed that the mutations of the ps5 stem contain a very high proportion of synonymous changes, such as mtDNA branches. In contrast, mutations in the outer pseudogene branches (Figure 1A, colored blue) contain a significantly higher proportion of non-synonymous changes (*p* < 0.00005), as expected for a truly pseudogenic, dysfunctional sequence. Thus, the ps5 sequence has been evolving under mitochondrial selective constraints, i.e., as a part of a functional mitochondrial genome, until it gave rise to a pseudogene, which then split into the *Homo*, *Pan*, and *Gorilla* variants. The impressive length of the ps5 stem implies that at the time of its insertion into the nuclear genome, the mtDNA predecessor of ps5 NUMT was highly divergent from the *Homo-Pan-Gorilla* ancestral mtDNA. Because the rate of evolution of mtDNA is relatively stable and well-documented, this divergence can be evaluated quantitatively with reasonable confidence.

### 2.2. The ps5 NUMT Should Have Been Transferred from a Separate Species 

A qualitative visual comparison shows that the length of the ps5 stem is comparable in length to the *Homo* and *Pan* mtDNA branches (Figure 1A). In other words, by the time ps5 was created, the mtDNA predecessor of the ps5 pseudogene should have diverged from the *Homo-Pan-Gorilla* mtDNA almost as far as the human and chimpanzee mtDNA diverged from each other. This suggests that the ps5 stem mtDNA lineage may represent a separate, now extinct, Hominine species. Because the ps5 sequences now reside in the *Homo*-*Pan*-*Gorilla* genomes, this extinct Hominine should have somehow transferred these sequences to the *Homo-Pan-Gorilla* clade, which implies a distant hybridization (Figure 1B). For a quantitative assessment, we estimated the mtDNA divergences within various hominine taxa and compared them to the divergence of the ps5 predecessor mtDNA using the maximum likelihood/jackknife approach (multiple resampling of the sequence shortened by removing 50% of the base pairs at random, Appendix A). We used a dataset of 82 great ape mitochondrial genomes [26], supplemented with human, Neanderthal, and Denisovan mtDNA. Importantly, this dataset was designed to represent the diversity of the great apes, as has been confirmed by the analysis of nuclear DNA of the same samples. Here we use the term “% divergence” to describe the divergence as inferred by the ML/Jackknife procedure. This measure is highly correlated with the widely accepted divergence *times* of the ape species (orangutan, gorilla, chimpanzee) and the gorilla and chimpanzee subspecies (Appendix A therein). The resulting distribution of jackknife estimates (Figure 2) shows that the ps5 stem branch (turquoise) diverged by 4.5 ± 0.8% from its common ancestor with *Homo-Pan-Gorilla* mtDNA by the time the pseudogene had been formed.

How much is a 4.5% mtDNA divergence from the *taxonomic* point of view? As seen in Figure 2, the estimates of mtDNA divergences between congeneric species (i.e., species belonging to the same genus, represented by the thin curves on the left) are very well separated from the divergences between genera (the thick purple, green, and grey curves on the right). The divergence of the Ps5 predecessor mtDNA is intermediate between the divergences of congeneric species and the divergences of genera. We thus conclude that the Ps5 precursor mtDNA and its host, the hypothetical extinct Hominine, belonged to a separate species, which significantly diverged from the *Homo-Pan-Gorilla* clade. Of note, the divergence between the great ape mtDNA sequences increased essentially linearly with the separation time between species at about 1% per 1 Myr (Appendix A). Thus, the extinct Hominine should have diverged by about 4.5 ± 0.8 Myr. Thus, 4.5 My is a time typically considered sufficient for significant isolation of the diverging species and such hybridization would seem impossible. However, reassuringly, a similar scenario, including the formation of a NUMT and its transfer to a divergent species by distant hybridization, has been recently described for Colobine monkeys [24] (see Discussion, Section 3.1).

### 2.3. Divergence of the ps5 Precursor mtDNA Cannot Be Explained by the Larger Size of the Ancestral Population

A potential alternative explanation of the high divergence of the mtDNA precursor of the ps5 pseudogene could be a highly effective population size of the mtDNA (**Ne_mit_**) of the ancestral population. In this case, the expected inter-individual genetic heterogeneity can be so large that a highly genetically divergent individual could have been merely a regular member of the population, rather than an intruder from a distant species. Thus, a potential limitation of our analysis is that we used present-day hominine populations as a reference to assess the divergence in an ancient population, whose effective size is generally believed to be larger than that of modern great ape populations. Therefore, we asked whether a larger effective size of the ancestral population (**Ne**) rather than the taxonomic distance of the ps5 carrier could have accounted for the surprisingly high apparent divergence of the ps5 precursor mtDNA. Of note, we need to distinguish between the nuclear DNA **Ne**, **Ne_nuc_**, and the mtDNA **Ne**, **Ne_mit_**. **Ne_nuc_** and **Ne_mit_** can be very different. Theoretically, **Ne_nuc_** is expected to be four times larger than **Ne_mit_**, but the ratio depends on the particular population dynamics.

The **Ne_nuc_** of the great ape ancestral populations has been recently estimated [26]. Although the *mitochondrial*
**Ne_mit_** of the ancestral population is not known, we can use modern effective population sizes of mtDNA and nuclear DNA in order to estimate their ratio (**Ne_nuc_**/**Ne_mit_**), and, assuming that this ratio is fairly stable across evolutionary time, infer the ancient **Ne_mit_**. Thus, we estimated (**Ne_mit_**) in the ancestral population in two steps. First, we determined how the mitochondrial **Ne_mit_** relates to the nuclear **Ne_nuc_** in modern hominine populations; second, we extrapolated that relationship to the ancestral population, assuming a constant **Ne_nuc_**/**Ne_mit_** ratio, and finally, used this ratio to calculate the estimated mitochondrial **Ne_mit_** of the ancestral population.

We first plotted the available data on the *maximum* mtDNA divergence within present-day chimpanzee and gorilla populations. As a proxy for “populations”, we used the formally accepted subspecies, conservatively assuming that individuals within a subspecies are sufficiently interconnected to be considered a population. The resulting plot (Appendix A) revealed a very weak, nonsignificant correlation between **Ne_nuc_** and the intraspecies mtDNA divergence. Linear extrapolation of these data to the higher **Ne_nuc_** of the ancestral population shows that the anticipated mtDNA divergence in the ancestral population should have been much lower than the divergence of the mtDNA precursor of the Ps5 pseudogene (Appendix A), in accordance with the “distant hybridization” hypothesis.

A possibility remains that the **Ne_nuc_**/**Ne_mit_** was not constant and mtDNA divergence of the ancestral population was higher relative to **Ne_nuc_** than that of modern populations. This would imply, however, that the ancestral population was structurally or otherwise significantly “different” from modern hominine populations in a way related to mtDNA or sex (Appendix A). For example, the male/female behavioral/migration patterns could have been different. Excessive divergence of mtDNA could potentially be explained by the relative immobility of females. These alternative possibilities, however, are perhaps even more peculiar and exciting than the hybridization scenario.

### 2.4. Interspecies Hybridization Was Not a Unique Event: Evidence from NUMTs ps11 and ps7

Ps5 is not the only mtDNA pseudogene that implies an interspecies hybridization event. Another pseudogene with a similar evolutionary history has been found on chromosome 11. Overall, the joint tree topology of the ps11 NUMT with mtDNA (Figure 3) is similar to that of ps5 (i.e., a long common pseudogene stem consisting of highly synonymous “mitochondrial” mutations and subsequent divergence among humans, chimpanzees, and gorillas). However, this pseudogene shows consistently higher similarity to the mtDNA of the gorilla than to that of other hominines (note a common stem segment with gorilla mtDNA in Figure 3). Note that in this case, the shape of the mtDNA sub-tree poorly reflects the evolutionary history of humans, chimpanzees, and gorillas. This is because NUMT ps11 is homologous to the rRNA section of the mitochondrial genome. This genome segment is known to poorly reflect evolutionary history, possibly because of excessive selection pressures.

It is tempting to speculate that the carrier of the Ps11 precursor mtDNA first belonged to the gorilla clade, then diverged into a separate lineage where it was inserted into the nuclear genome as a pseudogene, which was then transferred back to the gorillas as well as to the human/chimp clade by hybridization. Interestingly, a similar evolutionary scenario has been proposed based on the relationship between human and gorilla lice [27]. In that study, a gorilla-specific louse strain was shown to have been transferred to humans from gorillas 3.2 (+/−1.7) million years ago. Transfer of lice presumably requires close, persistent contact between members of the gorilla lineage and our ancestors. We cannot determine with certainty the time of the ps11 transfer to the human/chimp clade because ps11 is relatively short and does not afford estimates as precise as those for ps5, but it likely falls within the anticipated broad time range of the lice transfer. It is tempting to speculate that this contact resulted in the transfer, in addition to lice, of some genes (via inter-species hybridization) and that the pseudogene Ps11 is a relic of this (or similar) transfer. It is important to appreciate, however, that Ps11 mtDNA most likely has considerably diverged from gorillas by the time of becoming a pseudogene.

There is at least one more NUMT with a similar evolutionary history, which is located on chromosome 7 (Figure 4). This “ps7” NUMT is very old and diverged from our lineage around the time of the Afro-Eurasian monkey/ape separation. The corresponding “ps7 precursor” mtDNA has accumulated almost 9% nucleotide changes prior to its insertion into the nuclear genome. Using the same arguments as for the ps5 pseudogene, we conclude that, at a certain point, there was hybridization between species with mtDNA diverged by about 9% from their common ancestor. This is a very large divergence by modern ape standards, such as the divergence between orangutans and hominines (Figure 4). We will return to the plausibility of such hybridization in the Discussion.

## 3. Discussion

### 3.1. Are Such Distant Hybridizations Plausible? Comparison to Other Primates

The NUMT data strongly suggest that our direct ancestors were repeatedly involved in hybridization with distant species separated by about 4.5 Myr (ps5) and perhaps even prior (evidenced by ps7) to the hybridization event. Are such distant hybridizations at all possible? It is thought that within mammals, it takes around 2–4 Myr, on average, to establish reproductive isolation through hybrid unviability [28]. However, there is considerable variation across taxa. Natural hybridization has been estimated to occur between 7 and 10 percent of primate taxa [29]. Most of the evidence for primate hybridization is genomic, though some phenotypic studies have also been undertaken [30,31,32]. Although the bulk of hybrids are formed between congeneric species, more distantly related intergeneric primate hybrids do occur. For example, fertile intergeneric hybrids have been documented in crosses between baboon and gelada lineages, separated by circa 4 Myr [33]. Hybridization in captivity has also occurred between rhesus macaques and baboons (so-called rheboons), which diverged considerably further back in time, but the rheboons are not fertile [34]. There is also evidence that some living primate species are the products of hybridization [35,36,37,38]. In one case, the kipunji (*Rungwecebus kipunji*), a baboon-like monkey, appears to be the product of hybridization circa 600 Ka between taxa whose mtDNA lineages diverged 4–6 Ma, indicating that gene transfer occurred between 5.5 Myr and 3.5 Myr after the separation of the lineages [38]. Recently, evidence has appeared for hybridization between the Asian Colobine genera *Trachypithecus* and *Semnopithecus*, separated by the time of hybridization by about 5.5 Myr or more [24]. Intriguingly, in this case, the evidence of the hybridization is based on a NUMT present in *Semnopithecus*, which is closely related to the *Trachypithecus* mtDNA. This implies a scenario almost identical to what we have proposed for ps5 and other pseudogenes. Given this evidence, a separation time of about 4.5 Myr between the parties of the “Ps5 pseudogene transferring hybridization”, while very large, appears not unprecedented among primate lineages.

The hybridization implied by the ps7 data (9% divergence) appears too distant. It should be noted, however, that the ancestral population where this interbreeding would have taken place thrived about 30 million years ago and that little is known about its size and structure. Of note, most extant Afro-Eurasian monkeys practice male exogamy; if this were true for the ape/Afro-Eurasian monkey ancestral population, then this could have promoted high mtDNA divergence in a subpopulation, whose nDNA would not be so drastically diverged as it would be in a contemporary great ape population with the same mtDNA divergence, and thus still allowed for successful hybridization. An extreme example of such a situation is provided by the naked mole-rat, where mtDNA divergence even *within* a species with rather closely interrelated nDNA reaches as high as about 5%, presumably because of the extreme immobility of female queens in this eusocial rodent. The immobility of females results in an increased divergence of mtDNA because, in this situation, local mtDNA types are rarely replaced by types from distant areas of the same population and thus can accumulate more mutations. Even with these explanatory assumptions, the divergence of the ps7 pseudogene precursor is truly extraordinary. With these mtDNA pseudogenes as a lead, it would be interesting to look for other possible records (perhaps among nuclear loci) of distant interbreeding between the ancestors of our species.

### 3.2. Timing of Hybridization and the Fossil Evidence

Of particular interest is the timing of the asserted hybridization event. The phylogeny of the ps5 sequences consistently places the pseudogene insertion around the time of the *Homo/Pan* split, i.e., about 6 million years ago (Appendix A). In other words, the formation of the pseudogene and possibly the interspecies hybridization event took place within the Miocene epoch, when the ape lineages were diverging from each other, and the human lineage was diverging from the chimpanzee clade. Intriguingly, at the terminal part of the Miocene and the early Pliocene, certain hominin fossils have been interpreted alternately as more human-like or more ape-like in different respects. For example, there is considerable disagreement about the placement of *Sahelanthropus tchadensis* (7–6 Ma) on human versus ape lineages [39,40,41], in part because it combines achimp/bonobo-like cranial base and vault with more hominin-like traits, such as an anteriorly placed foramen magnum. Similarly, *Orrorin tugenensis* (~6 Ma) appears to have bipedal features of the femora [42,43,44], linking it to hominins but with ape-like dental morphology. Although a mixture of plesiomorphic and derived characters is to be expected in evolution, the mosaic nature of these taxa nonetheless makes them uneasy members of our clade. One possible interpretation of their taxonomic position is to place them in a separate clade of apes that shares convergent features (homoplasies) with the hominin clade [45]. However, it is also possible that some of these fossil specimens display mixed morphology as a result of genetic exchange between the ape and hominin lineages. If so, this would point to a complex process of lineage divergence and hybridization early in the evolution of our lineage, with the Ps5 pseudogene representing a genomic record of such a hybridization event.

Our estimate for the timing of the alleged ps5-related hybridization event based on mtDNA/NUMTs analysis coincides with that obtained by Patterson and colleagues using nuclear DNA data, i.e., “later than 6.3 Myr ago” [16]. We note, however, that these two methods do not necessarily detect the same events. As discussed below, interbreeding events might have been relatively common in the evolution of our species. This coincidence may therefore indicate that such events occurred more frequently at this critical time in our evolution, during the early divergence of the chimp/hominin lineages. Indeed, our preliminary data indicate that the formation of mtDNA pseudogenes appears to be punctuated, potentially correlated with the epochs of speciation in the hominin lineage [20]. Moreover, while our NUMT-based analysis documents gene exchange between genetically diverse individuals with fair certainty, it provides little information on the volume of the gene flow associated with the event. In principle, almost no genes other than ps5 itself might have been transferred from the putative extinct hominine to the *Homo-Pan-Gorilla* lineage. Therefore, it is possible that the interbreeding event recorded by the ps5 pseudogene might have gone undetected by the approach used by Patterson et al., while the event they describe might not have left any NUMT record detectable by us.

### 3.3. Multiple Pseudogenization/Hybridization Events: Potential Positive Selection of the Pseudogene?

It appears that the insertion of a pseudogene coincides with or was swiftly followed by the hybridization of distant lineages and was not a unique event in human evolutionary history. Although the consequences of hybridization vary widely, they can include the evolution of novel genotypes and phenotypes and even new species [45,46,47,48,49]. In the case of the human lineage, the adaptive fixation of introgressed genes appears to have occurred repeatedly, resulting in novel gene amalgamations that provided fitness advantages. For example, Neanderthal genes related to keratin production have been retained in populations living today [50,51]. Similarly, genes associated with immunity and adaptations to high-altitude environments in living people were acquired through ancient introgression [52]. It is, therefore, possible that the introgressed pseudogenes described here were linked to other genes that were themselves adaptively beneficial.

It is tempting to speculate on the possible mechanisms whereby these NUMTs got fixed in the population. Notably, the fixation process should have been rather efficient since these pseudogenes appear to have been fixed in more than one population. For example, ps5 was independently fixed in the gorilla and the human/chimp nascent populations, which by that time were probably substantially separated. This implies that the spread of the pseudogene within and across populations might have been driven by positive selection. Interestingly, indeed, both ps5 and ps11 are located close to the 3′ regions of functional genes (ps7 is yet to be studied in this respect). Insertion of an mtDNA pseudogene into the immediate vicinity of a functional gene is expected to be a strongly non-neutral mutation. In addition to a significant spatial disruption of the genome (e.g., ps5 was a 10 Kb+ insertion), the inserted mtDNA has very unusual properties, e.g., unprecedented strand asymmetry and potential for secondary structure formation (multiple RNA genes). Such an insertion is expected to significantly alter either the gene or its expression. Thus, pseudogene insertion should be either highly disruptive or, rarely, significantly beneficial. The mtDNA pseudogenes that remain after millions of years may represent those rare, significantly beneficial events.

Most intriguingly, ps11 is located within just a few hundred base pairs from the 3′ transcription termination site of the RNF141/ZNF230 gene, which is essential for spermatogenesis and fertility [53,54] It is worth noting that differences in the expression patterns of the RNF141 gene were proposed to contribute to the fast speciation of East African cichlids [55], especially in the context of their strong sexual selection. Thus, it is tempting to speculate that the insertion of the ps11 pseudogene served as an expression modifier for RNF141, which resulted in increased fertility and reproductive selective advantage and eventually allowed the pseudogene to spread over the human, chimpanzee, and gorilla ancestral populations.

Interestingly, RNF141 appears to be among a few genes demonstrating a selectively driven expression shift in the testis of the ancestor of hominines [56]. This phenomenon is perfectly in line with our hypothesis of adaptive fixation of pseudogene-induced changes in the expression level of the gene. It is important to emphasize the unique nature of this expression shift—except for the testis in the ancestor of hominines, RNF141 did not demonstrate any adaptive expression changes in seven other investigated tissues and all other branches of the mammalian phylogenetic tree [56]. This strongly supports the possibility that the gene went through a phase of pseudogene-induced intense selection during the speciation of hominines. 

### 3.4. Alternative Possibility: mtDNA Introgression

It should be noted that a similar NUMT genealogy would have been generated in a ‘symmetrical’ hybridization scenario, where a distant species donates its mtDNA rather than nuclear DNA loci, including the NUMT. Under this scenario, the NUMT is created in the acceptor species from its ‘old’ mtDNA, which later gets replaced in the population by the ‘new’ mtDNA of the introgression species. The schematic of this event is shown in Figure 5A, where it is assumed that introgression from a divergent species affects humans and chimpanzees only (HC-introgression). The latter condition is important because if introgression affected gorillas as well, the resulting shape of the mtDNA HCG tree, i.e., approximately equal H, C, and G branches, would have been incompatible with the observed (HC)G topology. As shown in the figure, either of the two scenarios results in a sub-tree of NUMTs with a long stem comprised of mitochondrial mutations, as is observed. The two scenarios are not fully symmetric, however. For example, the proposed mtDNA HC-introgression scenario is expected to result in a common stem between the gorilla branch and the NUMT sub-tree (see thick green arrows in Figure 5B). The length of this common gorilla/NUMT stem depends on how much time elapsed between the split of the extinct hominine and the insertion of the NUMT into the nuclear genome. Intriguingly, we indeed detect various degrees of affinity for the gorilla branch of both the ps11 and ps5 NUMTs. The significance of this preliminary observation, however, awaits further investigation. All NUMTs originating in the time interval from the split with the hypothetical extinct hominine till introgression should originate from the ‘old’ mtDNA.

## Figures and Tables

**Figure 1 genes-13-00810-f001:**
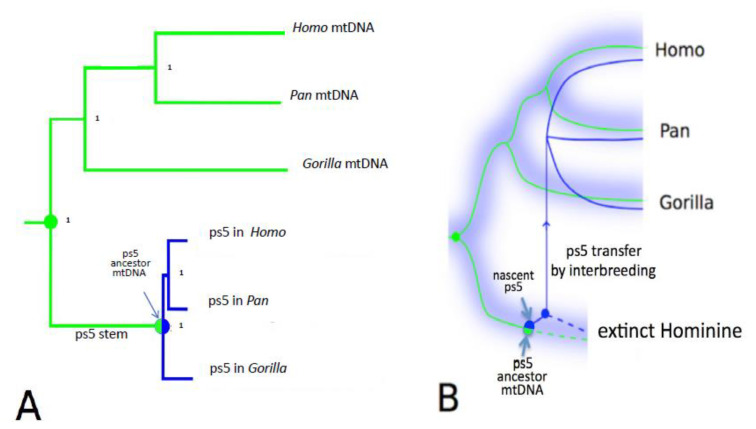
A joint phylogenetic tree of the hominine mtDNA and the ps5 pseudogene of mtDNA. (**A**) Green and blue lines depict the mitochondrial and the pseudogene lineages, respectively, diverging from their mitochondrial common ancestor (green circle). The common pseudogene stem (“Ps5 stem”) is colored green because, remarkably, mutations of the “ps5 stem” are mostly synonymous changes that must have occurred in a functional mitochondrial genome. This contrasts with a low fraction of synonymous changes in the pseudogene branches (blue). Note that the pseudogene branches are short because of the low mutation rates in nuclear DNA compared to mtDNA). The length of the “ps5 stem” implies ~4.5 My of evolution. The intriguing question is how did ps5 get back into the *Homo-Pan-Gorilla* clade after its precursor had been diverging from this clade for millions of years. Orangutan, gibbon, and baboon outgroups were omitted for simplicity (see Appendix A for tree building approach and stability analysis). (**B**) Interpretation of the mtDNA/ps5 tree of Figure 1A. Blue hazy branches represent species, rather than individual loci, and blue haze schematically symbolizes the superposition of the phylogenetic trajectories of all the nuclear genetic loci. Because the ps5 stem branch is essentially mitochondrial, it must have been evolving within a continuous maternal lineage, which, to accommodate the very long ps5 stem, should have been diverging for ~4.5 My. The long separation period implies that this maternal lineage was a part of a separate species. That species should have eventually gone extinct and is thus labeled “extinct Hominine”. The ps5 pseudogene was created in the extinct Hominine (half blue/half green circle) and transferred to the *Homo-Pan-Gorilla* clade via interspecies hybridization (thin blue arrow). Solid lines represent lineages currently extant lineages, dotted lines—extinct lineages.

**Figure 2 genes-13-00810-f002:**
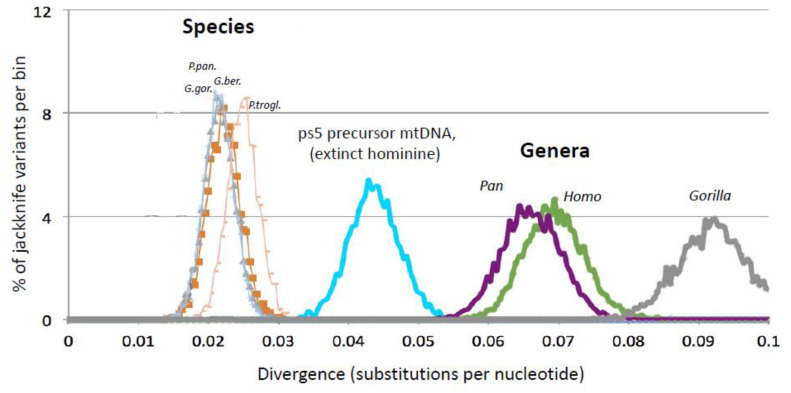
The divergence of hominine taxa from their common ancestors with sister taxa (a.k.a. branch lengths) is compared to the divergence of the ps5 precursor mtDNA (mtDNA of the hypothetical extinct Hominine) from its common ancestor with living hominines at the time of the ps5 formation (turquoise). Note that divergence of the ps5 precursor is intermediate between the divergences of congeneric species (*P. troglodytes* and *P. paniscus*; *G. gorilla* and *G. beringei*) and the divergences of genera (*Homo* and *Pan*). Divergences were estimated from the common ancestor with a sister taxon (e.g., for *Homo*—from the common ancestor with *Pan*, for *P. troglodytes*—from the common ancestor with *P. paniscus*, and for Ps5 precursor mtDNA—from the common ancestor with the *Homo-Pan-Gorilla* clade). The curves represent the distribution of estimates by the jackknife procedure (Appendix A); the ps5 data has been corrected by the fraction of mtDNA mutations in the ps5 stem (i.e., multiplied by 0.75, Appendix A).

**Figure 3 genes-13-00810-f003:**
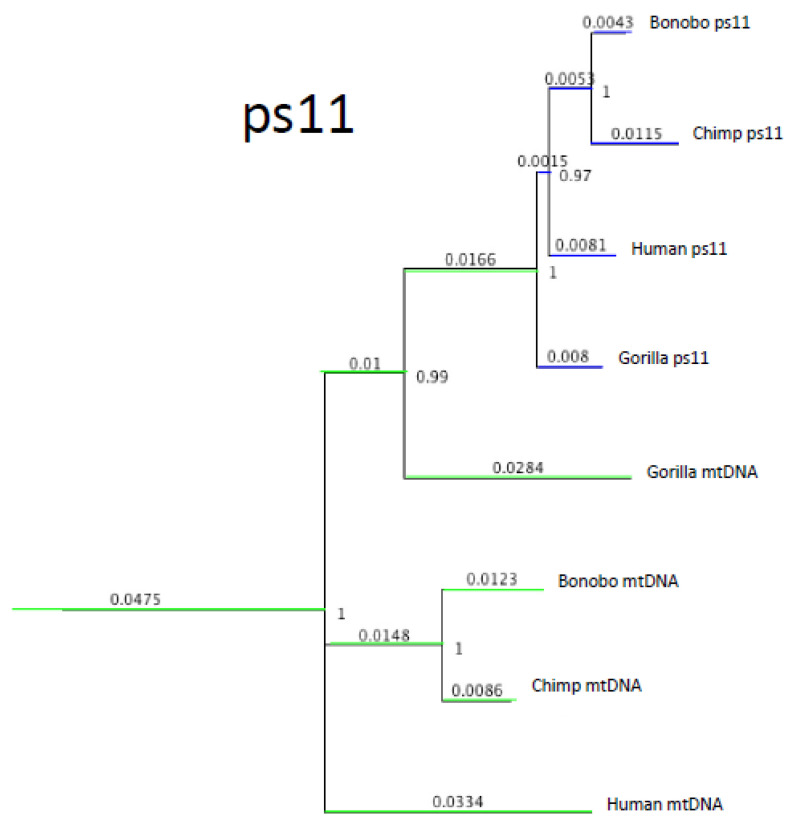
Phylogenetics of the pseudogene Ps11 and Hominine mtDNA. PhyML GTR. Note that this tree is still under construction; there are some unresolved problems in topology. In fact, this particular region of mtDNA is generally not apt for genealogical analysis because of high conservation and high selective pressures (as it is an rRNA coding region).

**Figure 4 genes-13-00810-f004:**
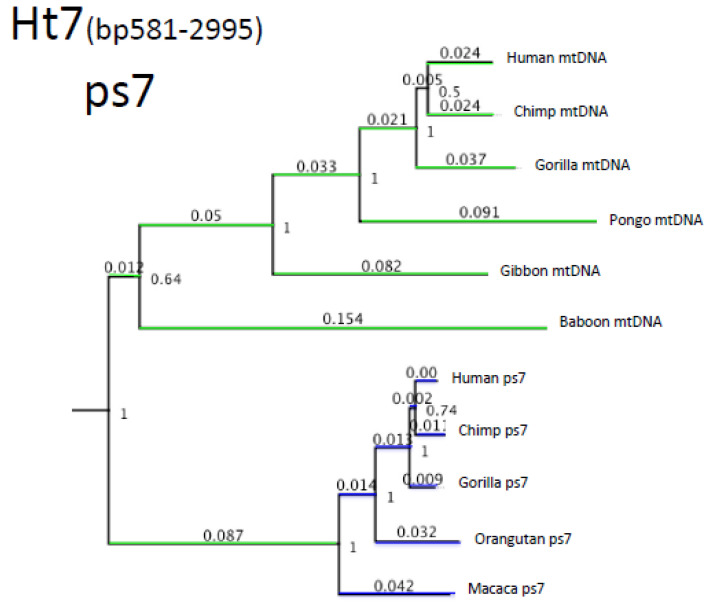
Phylogenetics of the pseudogene Ps7 and old-world monkey mtDNA.

**Figure 5 genes-13-00810-f005:**
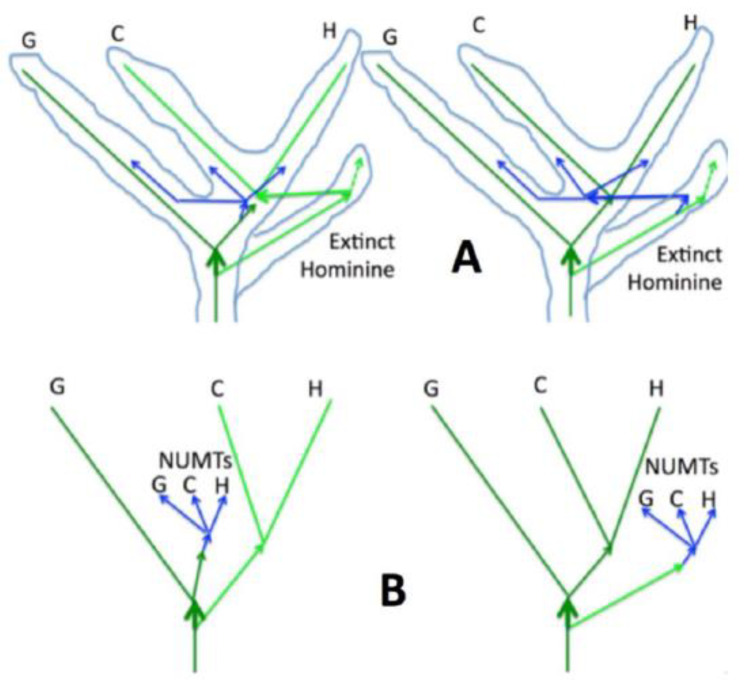
Comparison of the NUMT transfer (**right**) and the mtDNA introgression (**left**) scenarios of hybridization. (**A**) represents the inferred events, (**B**) represents the resulting observed phylogenies. Blue arrows represent the NUMTs, and light blue lines depict the superposition of the phylogenetic trajectories of all the nuclear genetic loci, as in Figure 1B. The mtDNA genealogies are green. Horizontal arrows represent introgression of mtDNA (green) or transfer of the NUMT locus between extinct hominine and the HCG mitochondrial (blue). Extinct hominine mtDNA lineage is neon green, the HCG mtDNA lineage is forest green.

## Data Availability

Not applicable.

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
