# Peer review of "Mitochondrial Pseudogenes Suggest Repeated Inter-Species Hybridization among Direct Human Ancestors"

_genes, 2022, doi:10.3390/genes13050810_

Round 1

Reviewer 1 Report

I have had good time at reading. I like the paper and how it is presented. The findings are novel and interesting and the authors did a clever job at stressing their findings. Some parts, though, are overemphasized or just too much inferential either. I do attach one of the comments in the edited pdf to make this point clearer:

the authors write:

"However, it is also possible that some of these fossil specimens display mixed morphology as a result of genetic exchange between the ape and hominin lineages. This would point to a complex process of lineage divergence and hybridization early in the evolution of our lineage, with the Ps5 pseudogene representing a genomic record of such a hybridization event."

to which I retort:

"this is vey much over the edge.

Sahelanthropus ok, it could well be on the Gorilla lineage, but Ardipithecus and any later australopiths nobody would doubt. Please do not overemphasize the mixture of plesiomorphic and derived characters as well. I may provide dozens of examples of the very same statement for members of the Homo lineage nobody would take as anything but Homo, including H. floresiensis, H. heidelbergensis, early H. sapiens, H. rudolfensis and many more. You don't have to call in hybridization for that, it's just evolution. I strongly, very strongly suggest you to tone down significantly this statement, and to review the argument that the phenotypic mixture of plesiomorphic and derived trait is anything we should be surprised about"

there are a couple of similar passages in the text I highlighted. The authors should fix that, after which this will look fine a contribution

regards

Author Response

We thank the reviewers for positive evaluation of the manuscript. We have addressed all reviewers’ comments, which helped to improve and update the manuscript.

A number of editorial changes were made throughout the manuscript for clarity and accuracy, consistent with Reviewer #2’s comments. We also made updates to the introduction to align it with the most recent literature, based on the comments of Reviewer #2.

Both of the reviewers found the section referring to the human fossil record to be overstated. Although we chose to leave it in, we took Reviewer #2’s advice by briefly qualifying our comments by stating that a mixture of plesiomorphic and derived traits are to be expected in evolving lineages. We also removed reference to Ardipithecus, as we agree with the reviewers that it is not a compelling argument for this species, but that the earlier hominins and especially Sahelanthropus are more compelling.

As per second reviewer’s request, we have also clarified our hypothesis that insertion of ps11 might have served as an expression modifier and fall under adaptive selection. We fully agree that this hypothesis is speculative, and we clearly present it as such. We have further clarified that this hypothesis pertains to the second pseudogene, ps11, not to the ps5.  We have chosen to leave that hypothesis in as we are confident that it benefits the reader by putting the insertion of mtDNA pseudogenes described herein in an evolution perspective. Finally, the intriguing finding that this gene indeed went through selectively driven expression shift at about the time where the insertion is expected to take place is a really beautiful coincidence truly worthy reporting.  

All changes can be  seen in the file with track changes enabled.

We are now supplying high resolution figures as separate files.

Reviewer 2 Report

I am very happy to see a manuscript like this one, it is a well crafted document on an interesting concept. I don't have any major comments but I do have two things I would like to see addressed in a minor revision. 

First, the figures need to be of higher quality. They appear blurred and sloppy in the document. 

Second, I think it is a bit of a stretch to bring in past hominin lineages into the discussion. I do not think we can make inferences about their contributions to our genomes considering ancient dna methods do not allow us to recover DNA older than 1 mya. 

Author Response

(The authors gave the same response as above.)

Round 2

Reviewer 1 Report

The authors did a good job at reviewing, the new version is more balanced and less naif in anthropological terms. Some points of interpretation are still over the edge, but I'm not inclined to force my opinion on the authors narrative. Therefore, I recommend publication

Author Response

We would like to thank the reviewer for their considerations and continued comments to improve this manuscript.